# Lost in *HELLS*: Disentangling the mystery of *SALNR* existence in senescence cellular models

**Arianna Consiglio**[1]⊚, **Marco Venturin**[2]⊚, **Sabrina Briguglio**[2], **Clara Rossi**[3], **Giorgio Grillo**[1], **Stefano Bellosta**[3], **Maria Grazia Cattaneo**[2], **Flavio Licciulli**[1], **Cristina Battaglia**[2]*

**1** Institute for Biomedical Technologies, National Research Council (ITB-CNR), Bari, Italy, **2** Department of Medical Biotechnology and Translational Medicine (BIOMETRA), Università degli Studi di Milano, Milano, Italy, **3** Department of Pharmacological and Biomolecular Sciences (DISFEB), Università degli Studi di Milano, Milano, Italy

⊚ These authors contributed equally to this work.
* cristina.battaglia@unimi.it

**Data Availability Statement:** All the relevant data are present in the manuscript and the Supporting information data can be found in UNIMI Dataverse [https://dataverse.unimi.it/dataverse/SALNR_

## Abstract

Long non-coding RNAs (lncRNAs) have emerged as key regulators of cellular senescence by transcriptionally and post-transcriptionally modulating the expression of many important genes involved in senescence-associated pathways and processes. Among the different lncRNAs associated to senescence, *Senescence Associated Long Non-coding RNA* (*SALNR*) was found to be down-regulated in different cellular models of senescence. Since its release in 2015, *SALNR* has not been annotated in any database or public repository, and no other experimental data have been published. The *SALNR* sequence is located on the long arm of chromosome 10, at band 10q23.33, and it overlaps the 3' end of the *HELLS* gene. This investigation helped to unravel the mystery of the existence of *SALNR* by analyzing publicly available short- and long-read RNA sequencing data sets and RT-PCR analysis in human tissues and cell lines. Additionally, the expression of *HELLS* has been studied in cellular models of replicative senescence, both *in silico* and *in vitro*. Our findings, while not supporting the actual existence of *SALNR* as an independent transcript in the analyzed experimental models, demonstrate the expression of a predicted *HELLS* isoform entirely covering the *SALNR* genomic region. Furthermore, we observed a strong down-regulation of *HELLS* in senescent cells versus proliferating cells, supporting its role in the senescence and aging process.

## Introduction

The progressive increase of the average human lifespan has drawn attention on the cellular and molecular determinants of aging, the time-dependent physiological impairment of cellular and tissue functions that has emerged as the main risk factor for several high prevalence pathologies (e.g., cancer, cardiovascular disorders, and neurodegenerative diseases), now referred to as "age-related disorders" [1]. Among the different hallmarks of aging, cellular

HELLS] and DOI [https://doi.org/10.13130/RD_UNIMI/JY8KN5]. Short-read RNA-seq mapping tracks on UCSC genome browser: https://genome.ucsc.edu/s/cnr.itb.ba/short-read_RNA-seq. Long-read RNA-seq mapping tracks on UCSC genome browser: https://genome.ucsc.edu/s/cnr.itb.ba/long-read_RNA-seq.

**Funding:** This work was supported by 'Piano di sostegno alla Ricerca 2020' Grant (Università degli Studi di Milano, award no. PSR2020_BATTAGLIA_LINEA_B to CB and MV; award no. PSR2020_CATTANEO_LINEA_C to MGC), 'NutrAge' Grant (CNR-FOE 2021, award no. DBA.AD005.225) to AC. Bioinformatics analysis has been executed on ICT infrastructure funded by CNR-BiOmics project (Italian MUR, PON R&I 2014-2020 D.D. n. 424 28/02/2018) to FL. The funders had no role in study design, data collection and analysis, decision to publish, or preparation of the manuscript.

**Competing interests:** The authors have declared that no competing interests exist.

senescence recapitulates many molecular features of aging and senescent cells were demonstrated to cause age-related phenotypes by accumulating throughout the body over time [2,3]. Cellular senescence is a highly heterogeneous and permanent state characterized by many cellular and molecular alterations such as permanent cell cycle arrest, senescence-associated secretory phenotype (SASP) that promotes the secretion of pro-inflammatory molecules, increased senescence-associated β-galactosidase (SA-β-gal) activity, altered cellular and nuclear morphology, induction of anti-apoptotic genes, DNA damage, epigenetic modifications, altered proteostasis, changes in cellular metabolism and accumulation of dysfunctional mitochondria [3,4]. The senescence state can be induced *in vitro* by prolonged cell culturing, as first described by Hayflick and Moorhead [5], or by other numerous stimuli among which DNA damage, oxidative stress, oncogene overexpression and mitochondrial dysfunction [2,3].

In the complex regulatory network orchestrating cellular senescence, long non-coding RNAs (lncRNAs), a large and heterogeneous class of non-coding RNAs longer than 200 bp, seem to play a key role by transcriptionally and post-transcriptionally modulating the expression of many important genes involved in the above mentioned senescence-associated pathways and processes, and often showing an altered expression during senescence and aging [6,7]. Among the different lncRNAs associated to senescence, *SALNR* ("*Senescence Associated Long Non-coding RNA*"), was first described by Wu et al. [8] by microarray experiments aimed at identifying deregulated lncRNAs during replicative senescence of lung fibroblasts. Authors pointed to *SALNR* as a consistently down-regulated lncRNA not only in replicative senescence but also in other cellular models of induced senescence. Moreover, they showed that *SALNR* regulates senescence by interacting with the NF90 RNA-binding protein: NF90 acts as repressor of senescence in the nucleus by inhibiting the biogenesis of senescence-inducing miRNAs through the binding to their corresponding pri-miRNAs. In senescent cells, NF90 translocates in the nucleolus, losing its inhibitory activity on miRNA biogenesis; in this scenario *SALNR*, located in the nucleus, would prevent NF90 translocation, preserving its repressive activity on senescence [8].

Despite these evidences, since its release in 2015, *SALNR* has not been annotated in any database or public repository and no other experimental data have been published. *SALNR* gene should be located in the long arm of chromosome 10, within the 10q23.33 band, overlapping the *HELLS* gene. *HELLS* gene encodes for a helicase belonging to the SNF2 family of proteins involved in lymphoid lineage cellular proliferation. In the last years, next-generation RNA sequencing (RNA-seq) approaches enabled the sequencing of protein-coding and non-coding RNAs together with the assessment of differential expression transcriptomic analysis in perturbed biological conditions. The upgrading of long-read RNA-seq platforms allowed the production of reads containing kilobase-sized RNA fragments, thus opening the way to tackle challenging regions of the human genome and to investigate RNA isoforms [9].

In this paper, we have tried to disentangle the mystery of *SALNR* existence by exploiting both *in silico* and *in vitro* approaches. We show that *SALNR* transcript is completely overlapped to a *HELLS* transcriptional isoform, the RefSeq transcript XR_007061960. We validated the existence of this *HELLS* isoform by analysis of publicly available RNA-seq datasets and by semi quantitative RT-PCR experiments, while there is no evidence of the existence of *SALNR* as separate, independent transcript in our experimental setting. Moreover, we found a strong down-regulation of *HELLS* in cellular models of replicative senescence.

## Materials and methods

### Identification of *SALNR* genomic location and annotation

*SALNR* was identified in 2015 [8] as cDNA clone AK091544 and its location on the human genome was situated on the long arm of chromosome 10, on the Crick strand, chr10:

96353372–96356258 on NCBI36/hg18. We first converted its genomic location on the GRCh38/hg38 assembly on chromosome 10, chr10: 94603625–94606511. Then we explored both the Ensembl Release 108 (Oct 22) [https://www.ensembl.org/index.html] and NCBI reference genome and transcriptome [NCBI Homo sapiens Annotation Release 110 (2022-04-12)] to extract its putative sequence and the annotated portions of DNA overlapping or flanking its transcription area. To search for lncRNA annotations, we also checked LNCipedia v.5.2, a comprehensive compendium of human long non-coding RNAs [10] [https://lncipedia.org/].

## Search for *SALNR/HELLS* transcripts in public RNA-seq datasets

As first instance, we have searched for RNA-seq public datasets with accessible FASTQ files proving the transcription of *SALNR/HELLS* sequences and focusing on the study of replicative senescence in open repositories such as Gene Expression Omnibus (GEO, https://www.ncbi.nlm.nih.gov/geo/) and NCBI Sequence Read Archive (SRA, https://www.ncbi.nlm.nih.gov/sra). Overall, we selected five RNA-seq datasets from projects that compare senescent and proliferative human cell line experiments, listed in Table 1, including one dataset comparing proliferative and senescent WI-38 lung fibroblasts.

Moreover, to deeply investigate the structure of the genomic region located on chromosome 10 (chr10: 94603625–94606511 on GRCh38/hg38) including *HELLS* and *SALNR* transcripts, we checked for availability of FASTQ files of long-reads RNA-seq runs focusing on normal human tissues or cell lines in SRA.

To get comprehensive annotation on *HELLS*, three open resources have been consulted: Human Protein Atlas (https://www.proteinatlas.org/, Version: 22.0. Atlas updated: 2022-12-07), GTEx (https://www.gtexportal.org/home/, v8, update 2022-07-21) and GenAge (The Ageing Gene Database, https://genomics.senescence.info/).

## Bioinformatic analysis

RNA-seq FASTQ files downloaded from SRA were quality checked with FastQC tool [17] (see Data availability for FastQC output files). Both short (Table 1) and long read datasets (S1 Table) were mapped against the Ensembl genome and transcriptome (Release 108). Short

**Table 1. Senescence RNA-seq datasets used for differential expression analysis of *HELLS* gene and isoforms (short-read).**

| GEO accession SRA study | SRA run accession | Cell type | Condition | Read type | Reference |
|---|---|---|---|---|---|
| GSE155680 SRP275809 | SRR12385885, SRR12385886, SRR12385887, SRR12385888, SRR12385889, SRR12385890 | Human umbilical vein endothelial cells (HUVECs) | 3 senescent vs 3 proliferative | 60bp Illumina RNA-seq | [11] |
| GSE157867 SRP282193 | SRR13336755, SRR13336757, SRR13336758, SRR13336760 | HUVECs | 2 senescent vs 2 proliferative | 2x150bp (paired) Illumina RNA-seq | [12] |
| GSE163251 SRP298025 | SRR13255939, SRR13255942, SRR13255943, SRR13255946 | HUVECs | 2 senescent vs 2 proliferative | 2x50bp (paired) Illumina RNA-seq | [13] |
| GSE171663 SRP313881 | SRR14168768, SRR14168771, SRR14168772, SRR14168775, SRR14168776, SRR14168779, SRR14168780, SRR14168783 | Vascular smooth muscle cells (VSMCs) | 4 senescent vs 4 proliferative | 76bp Illumina RNA-seq | [14] |
| GSE63577 SRP050179 | SRR1660537, SRR1660538, SRR1660539, SRR1660540, SRR1660541, SRR1660542; SRR1660543, SRR1660544, SRR1660545, SRR1660546, SRR1660547, SRR1660548; SRR1660549, SRR1660550, SRR1660551, SRR1660552, SRR1660553, SRR1660554; SRR1660555, SRR1660556, SRR1660557, SRR1660558, SRR1660559, SRR1660560 | Human fibroblasts (4 types: BJ, HFF, IMR-90, WI-38) | 3 senescent vs 3 proliferative, for each cell type (12 samples) | 50 bp Illumina RNA-seq | [15,16] |

reads (50-150bp) were mapped with STAR [18], the expressions were estimated by RSEM v1.3.3 [19] and MultiDEA v1.1.0 [20] in order to manage multiread issues. Differential expression analysis of senescent and proliferative samples was performed with DESeq2 v1.36.0 [21], and the results were filtered by mean read count >50 for genes and >20 for isoforms, absolute log2 Fold Change (log2FC) >1 and adjusted p-value (FDR) <0.05. Long-read FASTQ files were mapped with deSALT v1.5.6 [22].

Below is a list of non-default parameters used for each tool. STAR was run through RSEM with—*star* option, and—*paired-end* was used for paired end samples, while deSALT was run with—*gtf* to use GTF annotation files and with *-x clr* for PacBio reads (PacBio SMRT CLR reads, error rate 15%) or *-x ont1d* for Nanopore reads (Oxford Nanopore 1D reads, error rate > 20%). A table containing read ID, transcript ID and mapping quality was extracted from the BAM output of mapping and used as input for MultiDEA's module *mapping_output_2_trapezoids.sh*. DESeq2 was applied to raw read counts by testing *design = ~ condition*, where condition is "proliferative" or "senescent". We applied DESeq2 at gene level to RSEM read counts and at isoform level to both RSEM and MultiDEA read counts, with the aim of correlating the differential expression events to the different isoforms of HELLS.

Since *SALNR* transcript totally or partially overlaps *HELLS* transcripts, some ad-hoc Bash scripts were developed to explore in deeper detail the characteristics of the reads mapping to their genomic location.

The results were loaded into the UCSC genome browser [https://genome.ucsc.edu/], and the corresponding URLs are listed in the Data availability section.

## Human cell lines

Normal Human Dermal Fibroblasts (NHDF) were purchased from Sigma (C-12302, Sigma). Cells were cultured in EMEM (ECB2071L, Euroclone), 10% FBS (ECS0180L, Euroclone), 1% NEAA (ECB3054D, Euroclone), 1% 200 mM glutamine (ECB3000D, Euroclone), 1% Penicillin/Streptomycin (ECB3001D, Euroclone). Human aortic smooth muscle cells (HSMC) were purchased from the American Tissue Culture Collection (PCS-100-012, ATCC, Manassas, USA). Cells were cultured in ATCC Vascular Cell Basal Medium (PCS-100-030, ATCC; 500 mL added with 500 μL ascorbic acid, 500 μL rh EGF, 500 μL rh insulin and rh FGF-b, 25 mL glutamine), 5% FBS (ATCC Vascular Smooth Muscle Growth Kit), and 5 mL Penicillin-Streptomycin 100X (Euroclone, Milan, Italy). Human umbilical vein endothelial cells (HUVEC) were purchased from Lonza (Bend, OR, USA) and routinely grown in Endothelial Growth Medium (EGM™-2) as indicated by the provider. The neuroblastoma cell line SH-SY5Y was purchased from Sigma (ECACC 94030304, Sigma) and cultured in EMEM (ECB2071L, Euroclone) and HAM'S F-12 (ECB7502L, Euroclone) 1:1, 10% FBS (ECS0180L, Euroclone), 1% NEAA (ECB3054D, Euroclone), 1% 200 mM glutamine (ECB3000D, Euroclone), 1% Penicillin/Streptomycin (ECB3001D, Euroclone). All cultures were maintained at 37˚C in a 5% $CO_2$ incubator.

## *In vitro* cellular models of replicative senescence

We set up the replicative senescence (RS) by prolonged culturing of NHDF and HSMC cells. We considered as young the cells at the sixth passage corresponding to Population Doubling Level (PDL) 20 and as old/senescent the cells at sixteenth passage corresponding to PDL60-65. HSMCs were also used at the sixth passage as young and at sixteenth as old. The senescence condition in the two cellular models was assessed by monitoring the morphological changes and by β-galactosidase (β-gal) staining with the Senescence Cells Histochemical Staining Kit (CS0030, Sigma-Aldrich). After reaching the preferred condition, cells were collected in TRIzol reagent (15596018, Invitrogen) and stored at -80˚C.

## RNA isolation from human cell lines

Total RNA was isolated from different human cell lines using TRIzol reagent according to the manufacturer's instructions. Concentration and purity of RNA were measured using the Nanodrop 1000 spectrophotometer (ThermoFisher Scientific). All RNA samples had an A260/280 value of 1.8–2.1. The quality of RNA was also evaluated using the Bioanalyzer 2100 instrument (G2939BA, Agilent) or Tape Station 2200 (Agilent). All the samples had a RIN value $\geq$ 7.

## RT-PCR and Sanger sequencing

RNA isolated from cells (1 μg) was digested using the RQ1 RNase-Free Dnase (M6101, Promega) and reverse-transcribed using the High-Capacity cDNA Reverse Transcription Kit (4368814, Applied Biosystems), according to the manufacturer's instructions. High-quality RNA from 12 human tissues (bladder, brain, cervix, esophagus, kidney, liver, lung, ovary, skeletal muscle, small intestine, spleen, and testis; First Choice Human Total RNA Survey Panel, AM6000, Ambion) was reverse-transcribed using Maxima H Minus cDNA Synthesis Master Mix with dsDNase kit (M1682; Thermo Fisher Scientific), according to the manufacturer's protocol.

RT-PCR was performed using the GoTaq G2 DNA Polymerase (M7845, Promega) with 3 μl of reverse transcriptase product and the following primers: for HELLS_XR_007061960 amplicon (811 bp) fw: TTTCAAAGGTGGTCAGTCTGGA; rev: TCCTACCGACAGCCAAGTCT; for HELLS_XR_007061960_3' amplicon (906 bp) fw: GGCTTTTCATGGGGAGAGATTG; rev: ATCACGAGTTTCAAGCAAGCCTA. PCR fragments were analyzed by gel electrophoresis with 1.8% agarose gels in TBE buffer containing 20000x SYBR Safe (S33102, Invitrogen). 5 μl of PCR products were loaded in each well. The bands were visualized using the Uvitec transilluminator (Cambridge). The two PCR products were directly sequenced in both directions using the Big Dye Terminator Cycle Sequence Ready Reaction Kit v3.1 (4337455, Applied Biosystems) and resolved on a 3130xl Genetic Analyzer (Life Technologies). The electropherograms were analyzed by means of the ChromasPro software (v2.1.5) (Technelysium Pty Ltd).

## Quantitative RT-PCR assays

Quantitative RT-PCR (qPCR) was performed with the QuantStudio 5 thermocycler (Applied Biosystems) in 384-well plates using the GoTaq qPCR Master Mix (A6002, Promega). Primers were designed by the Primer-BLAST tool (https://www.ncbi.nlm.nih.gov/tools/primer-blast/index.cgi) (S2 Table). A total of 10 μl PCR reactions were prepared containing 2 μl of reverse transcriptase product and 0.2 μl of each primer (10 μM). The PCR mixtures were incubated at 95˚C for 3 min, followed by 40 cycles of 95˚C for 10 s and 60˚C for 1 m. The results were analyzed with the QuantStudio Design & Analysis Software v1.5.2 (Applied Biosystems). All the experiments were performed at least in triplicate and the calculation of gene expression levels was based on the 2−ΔΔCt method. The geometric mean of the expression values of *CYC1*, *EIF4A2* and *RPSA* was taken as normalizer gene set; then the fold change obtained as ratio of normalized expression levels in old/senescent cells compared to young cells was calculated. Descriptive statistics, unpaired t-test and regression analysis of qPCR data were carried out by Microsoft Excel and/or by using Prism 8 (GraphPad Software, La Jolla, CA, United States).

## Results

### *In silico* analysis reveals the overlapping of *SALNR*/*AK091544* and *HELLS* isoforms

The *SALNR* lncRNA was published and characterized by Wu et al. in their 2015 paper [8]. The cDNA clone code corresponding to *SALNR* is AK091544, located on chromosome 10 (chr10:

94603625–94606511 on GRCh38/hg38), later extended via a 5'-RACE from which they saw it is actually longer at 5' end (chr10: 94602724–94606511). Since its publication, *SALNR* has been repeatedly cited among human lncRNAs, but its actual origin and transcriptomic products have not been further confirmed. Currently, the only annotations overlapping this genomic location are those of some transcripts of the *HELLS* gene (Ensembl ID: ENSG00000119969, Entrez Gene ID: 3070), located upstream on the same strand. Several *HELLS* isoforms/transcripts have been annotated in genomic databases: specifically, 21 isoforms in Ensembl GEN-CODE v42 annotation and 11 isoforms in NCBI RefSeq gene annotation release 110. Fig 1 shows the overlap among *SALNR*/AK091544 and *HELLS* isoforms. In particular, we found an overlap in the 3'-UTR portion of two Ensembl transcripts (ENST00000698800 and ENST00000419900), in the introns of ENST00000371327 and of ENST00000432120 lncRNA (ENSG00000244332 gene) and in an exon of a predicted RefSeq transcript (XR_007061960). Noteworthy, the RefSeq transcript XR_007061960 was added to *HELLS* RefSeq isoforms in April 2022 (NCBI Homo Sapiens Annotation Release 110, 2022-04-12) and it completely covers *SALNR* genomic location. Looking at the chr10: 94603625–94606511 genomic region (GRCh38/hg38) in the LNCipedia database we only retrieved the ENST00000432120 transcript (referred to LNCipedia ID: lnc-TBC1D12-1:1, https://lncipedia.org/db/transcript/lnc-TBC1D12-1:1), but no further annotation was available.

RNA-seq data collected from public repositories were mapped against the Ensembl human genome and the results were loaded into the UCSC genome browser (genome release GRCh38/hg38). Short-reads revealed an unexpected read coverage over the 3' end of *HELLS* gene, with a considerable number of reads mapped on *SALNR*/AK09154 region (see some examples in S1 Fig, and see Data availability section for online URL). Furthermore, long-read datasets were used to search for transcribed RNAs overlapping *SALNR*/AK09154 location as well as to identify the isoforms transcribed in the selected samples (S1 Table).

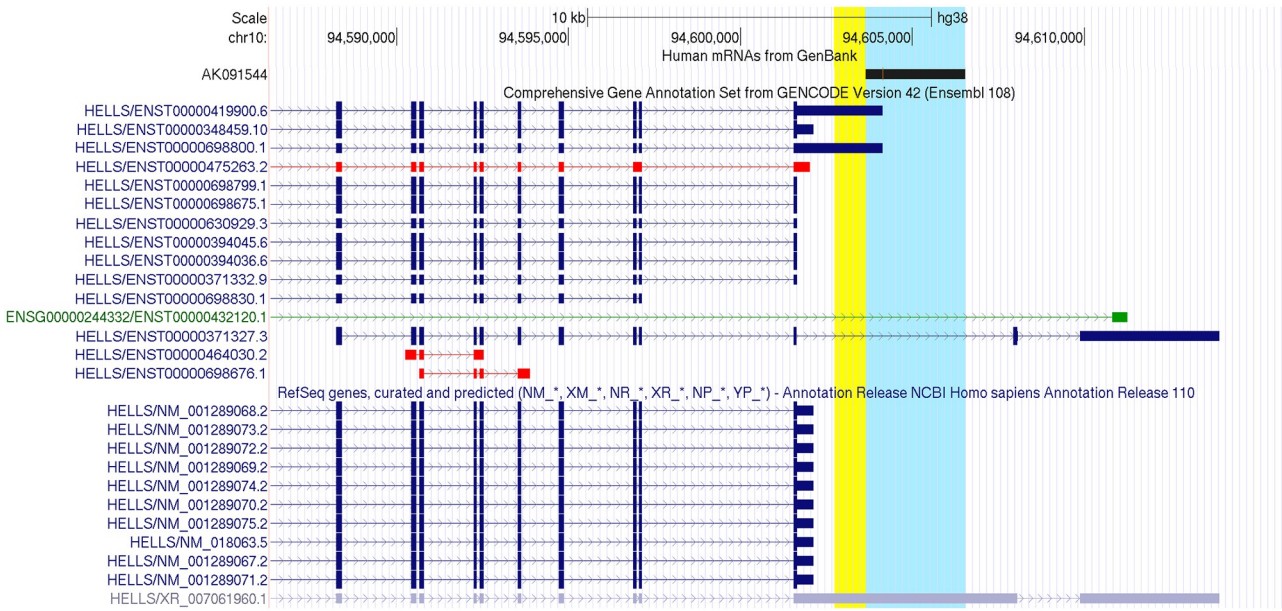

**Fig 1. Genomic location of *HELLS* and *SALNR/AK091544*.** A screenshot from UCSC genome browser showing AK091544 clone (highlighted in cyan, while the 5'-RACE result is highlighted in yellow) and the annotated transcripts of *HELLS* overlapping or flanking it from Ensembl GENCODE (only 14 out of 21 isoforms are included in this partial view) and NCBI RefSeq. GENCODE color legend: Blue, coding; green, non-coding; red, problem. NCBI RefSeq color legend: Blue, reviewed; lilac, predicted.

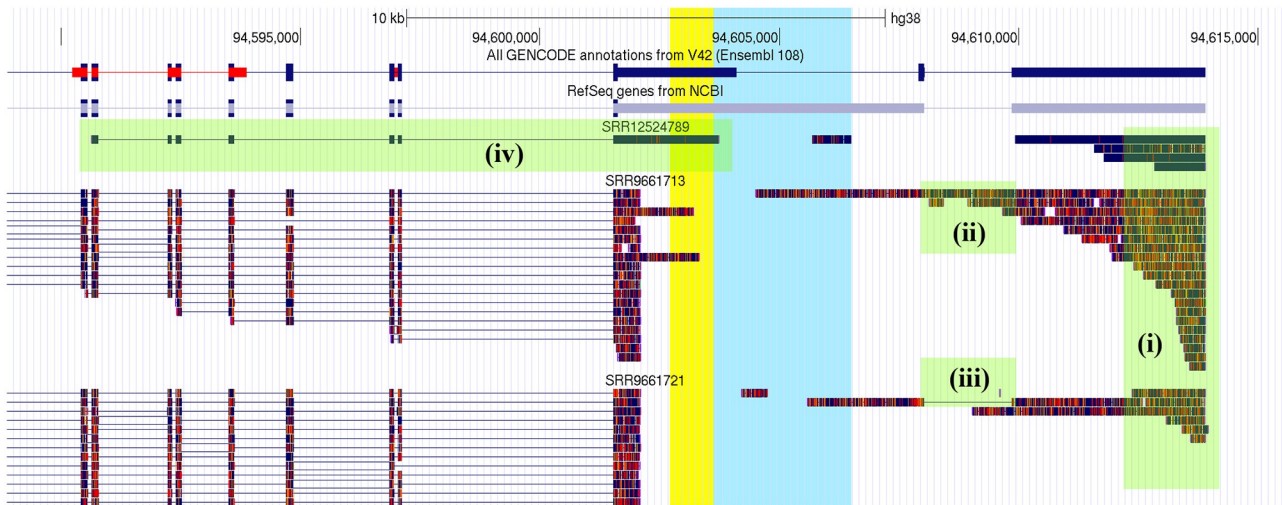

**Fig 2. Alignment of long–read RNA-seq data on *HELLS* genomic location.** The figure shows 3 representative samples of long-read RNA-seq and their alignment on *HELLS* genomic location. *SALNR*/AK091544 location is highlighted in cyan and yellow. Green boxes highlight that (i) reads mapping to the 3' end of *HELLS* gene confirm the end of the gene and the transcription of such portion; (ii) reads including the last intron of RefSeq XR_007061960 (see Fig 1) suggest the presence of another isoform and (iii) the reads excluding such intron confirm XR_007061960 3' configuration; (iv) the read in SRR12524789 run from SRP278890 confirms the presence of an isoforms that extends the 3' of an *HELLS* transcript over *SALNR*/AK091544 location. For the other samples see Data availability section.

In general, we found that long-read datasets showed poor coverage of *HELLS* genomic portion, and its transcripts are too long to be entirely represented, even by long-read sequencing (see Data availability section for online URL of long-read samples). However, as depicted in Fig 2, some interesting results are visible in the samples. Overall we observe that: the reads mapping to the 3' end of *HELLS* gene confirm the presence of at least one isoform that includes that portion, and the 3' end of XR_007061960 and ENST00000371327 is confirmed to be the definitive limit for the transcription of that genomic locus (chr10, position 94613905) (Fig 2.i); the reads including the last intron of XR_007061960 suggest the presence of another isoform (Fig 2.ii and 2.iii) the reads excluding such intron confirm XR_007061960 3' configuration (Fig 2.iii); the read in SRR12524789 run from SRP278890 confirms the presence of an isoform that extends the 3' of an *HELLS* transcript over *SALNR* location (Fig 2.iv); none of the reads exactly matches the *SALNR*/AK091544 location starting and ending in its candidate boundaries.

## RT-PCR confirms the presence of the predicted isoform of *HELLS* in human cell lines and tissues

Working with *HELLS* gene can be challenging because of its low expression in human tissues as depicted by the Human Protein Atlas (https://www.proteinatlas.org/ENSG00000119969-HELLS). Moreover, as shown by the GTEx database, the data availability of the expression levels of *HELLS* isoforms in human samples is limited to few of them (https://gtexportal.org/home/gene/HELLS). Therefore, to confirm the existence of the predicted *HELLS* isoform that completely overlaps the *SALNR*/AK091544 sequence, we designed primers that specifically amplify the RefSeq XR_007061960 predicted isoform (HELLS_XR_007061960 and HELLS_XR_007061960_3') (S2 Fig) and carried out reverse-transcription PCR (RT-PCR) on RNA extracted from several human cell lines (NHDF, HSMC, HUVEC, SH-SY5Y) and from 12 human tissues (see methods section). The semi-

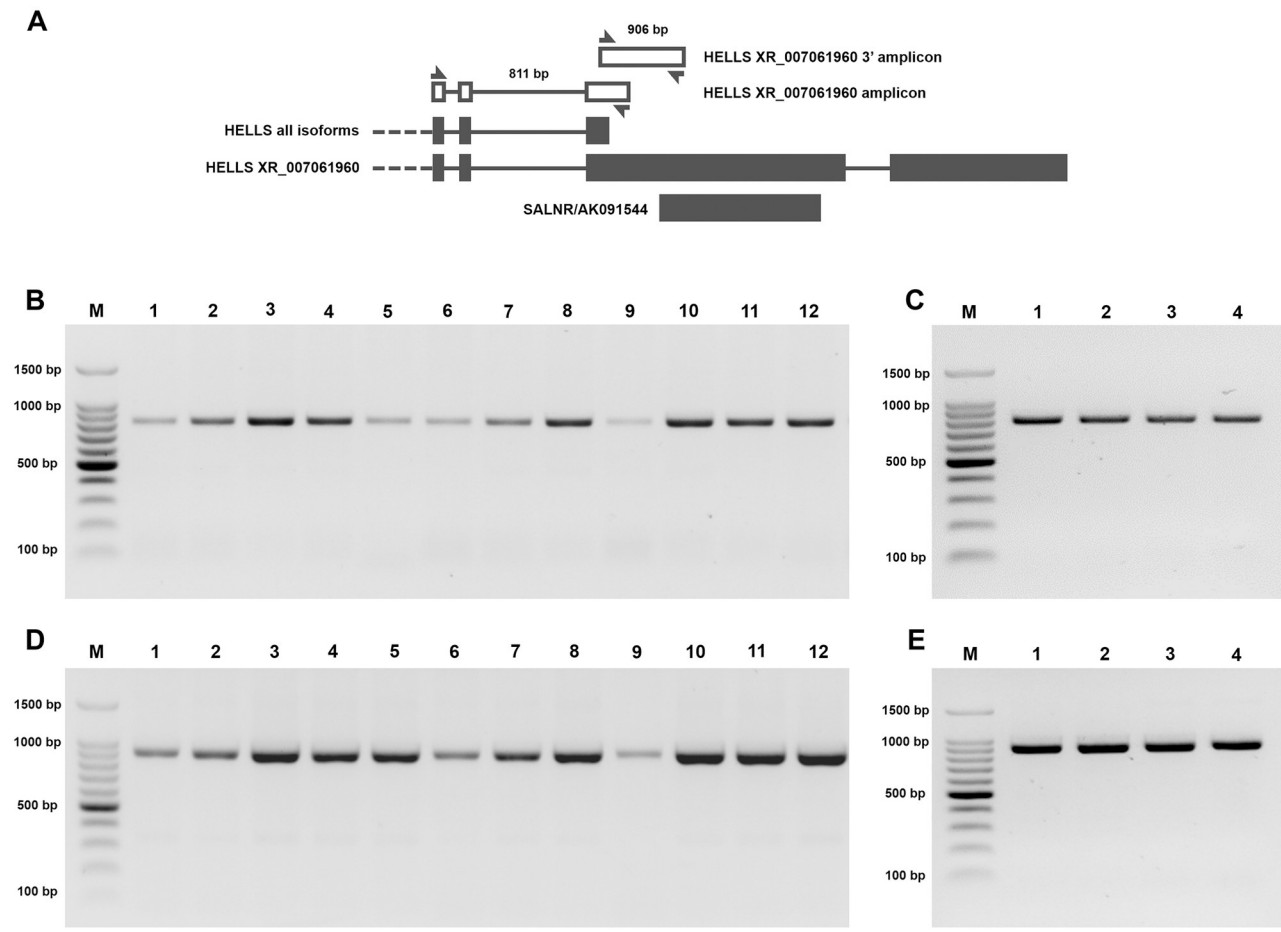

**Fig 3. Analysis of RT-PCR products specific for the *HELLS* XR_007061960 isoform.** The figure shows the gel electrophoresis analysis of RT-PCR products specific for the *HELLS* XR_007061960 isoform in human tissues and cell lines. A, scheme of the primers designed to amplify the *HELLS* XR_007061960 isoform and their genomic location in *HELLS* locus (see methods); B-C: HELLS_XR_007061960 amplicon, 811 bp; D-E: HELLS_XR_007061960_3' amplicon, 906 bp. PCR products were analyzed through a 1,8% gel in TBE buffer. Panel B, D: M. MW Marker, 1. Bladder, 2. Brain, 3. Cervix, 4. Esophagus, 5. Kidney, 6. Liver, 7. Lung, 8. Ovary, 9. Skeletal muscle, 10. Small intestine, 11. Spleen, 12. Testis. Panel C,E: M. MW Marker, 1. SH-SY5Y, 2. HSMC, 3. NHDF, 4. HUVEC.

quantitative RT-PCR analysis yielded in all cases a specific amplification product of the expected size (Fig 3). Sanger sequencing carried out on two representative RT-PCR products obtained from HSMC and SH-SY5Y cells confirmed the correspondence between the sequence of the amplicons and the one of the predicted isoform (S1 File).

### *In silico* differential expression analysis reveals under-expression of *HELLS* transcripts in senescence cellular models

Wu et al. observed that *SALNR*/AK091544 amplicon shows a reduced expression in cellular senescence [8]. Since *SALNR*/AK091544 genomic location is overlapped to the actual annotations of *HELLS* gene, we have performed a differential expression analysis of the RNA-seq reads expressed by the genomic location that included *HELLS* and *SALNR*/AK091544, on the datasets listed in Table 1, representing cellular models of replicative senescence. Since Ensembl and RefSeq annotations have a few different transcripts, we defined a single annotation for *HELLS* by manually adding the XR_007061960 sequence to the Ensembl GTF annotation file.

**Table 2. Differential expression analysis results for *HELLS* gene (ENSG00000119969).**

| GEO accession | Proliferative cells (read counts) | Senescent cells (read counts) | Mean read count | Log2 FC | adj. p-value (FDR) | Stat. sign. |
|---|---|---|---|---|---|---|
| GSE155680 | 823.9, 837.3, 1026.6 | 224.2, 256.1, 264.3 | 572.1 | -1.85 | 8.16E-50 | * |
| GSE157867 | 1451.2, 1515.3 | 814.4, 1043.8 | 1206.2 | -0.67 | 0.1274 | |
| GSE163251 | 1422.4, 4316.4 | 305.2, 379.6 | 1605.9 | -3.07 | 2.05E-06 | * |
| GSE171663 | 207, 237.1, 255.2, 382.1 | 53, 64.7, 104.9, 220.6 | 190.6 | -1.28 | 0.0625 | |
| GSE63577_BJ | 1252.2, 1254.8, 1375.4 | 703.6, 791.4, 805.5 | 1030.5 | -0.75 | 7.7E-19 | |
| GSE63577_HFF | 851.8, 867.8, 904.4 | 180.7, 193.4, 208.5 | 534.4 | -2.17 | 5.26E-86 | * |
| GSE63577_IMR90 | 1347.2, 1440.6, 1465.9 | 121.4, 153.2, 1473.8 | 1000.4 | -1.28 | 1 | |
| GSE63577_WI38 | 1315.6, 1442.5, 1449 | 263, 266.4, 269.7 | 834.3 | -2.40 | 5.64E-145 | * |

*SALNR*/AK091544 is absent in both the annotations and we decided to not consider it in this step. Firstly, we mapped the dataset with STAR and estimated *HELLS* expression (at the gene level) with RSEM. Then, we performed differential expression analysis with DESeq2 (Table 2), observing a decreased read count in senescent cells for each dataset. The statistical analysis resulted in a fairly reproducible down-regulation of *HELLS* at gene level, and in four out of eight datasets the Log2FC values reached the statistical significance (Table 2).

At this point, to isolate the source of under-expression events, we tried to distinguish the expression of the different *HELLS* isoforms. We analyzed a total of 23 isoforms: 21 Ensembl *HELLS* isoforms, ENSG00000244332/ENST00000432120 and RefSeq XR_007061960. Specifically, the *SALNR* genomic locus overlapped three isoforms of *HELLS*. Due to the evident overlap between transcripts, RSEM failed in distributing the reads among the different isoforms, because some of them turned out to have null expression even though there are reads mapping to their transcripts. MultiDEA provided a more detailed picture of the ambiguity in the mapping, and the only isoforms that achieve a significant number of uniquely mapping reads (>20) are XR_007061960 and ENST00000419900 (S3 Table). Moreover, the differential expression analysis performed on both RSEM and MultiDEA isoform expression estimations showed that under-expression events are not limited to those *HELLS* isoforms overlapping with *SALNR* genomic area (i.e., on ENST00000371327, ENST00000698800, ENST00000419900 and XR_007061960) but they involve all *HELLS* transcripts (S4 Table). This result contributes to question the existence of *SALNR* as an independent gene and to confirm the involvement of *HELLS* in senescence.

## qPCR confirms the down-regulated expression of *HELLS* in senescence cellular models

To verify the results obtained from *in silico* analyses, we evaluated the expression levels of *HELLS* during senescence by qPCR. We exploited two different senescence cellular models, NHDF and HSMC cells in replicative senescence, i.e., serially cultured until replicative exhaustion (passage p16). The senescent condition was validated by morphological changes (cellular enlargement and flattening with a concomitant increase in the size of the nucleus) and positive β-gal staining (S3 Fig). In addition, we confirmed their senescent status by detecting the gene expression levels of well-known markers of senescence in the old cells as compared to the young cells by qPCR. Overall, we observed in both cell lines the up-regulation of *CDKN1A/p21*, *CXCL8* and *MMP3*, and the down-regulation of *HGMB1* and *LMNB1* in the old cells compared to the young cells (Table 3), consistently with their senescent status.

Focusing on *HELLS*/*SALNR* expression, we designed two set of qPCR primers (S2 Table): the first one was able to amplify a consensus region shared by all *HELLS* isoforms, while the

**Table 3. Relative gene expression (fold change) for senescence markers and *HELLS* in NHDF and HSMC senescent cells compared to young cells obtained by qPCR.** ([a]fold change obtained as ratio of expression levels in old senescent cells compared to young cells; [b]SALNR amplicon overlapped with the XR_007061960 *HELLS* isoform).

| Samples | *p16* | *p21* | *CXCL8* | *GLB1* | *HMGB1* | *LMNB1* | *MMP3* | *HELLS* | *SALNR*[b] |
|---|---|---|---|---|---|---|---|---|---|
| NHDF R1[a] | 0.81 | 3.13 | 1.86 | 1.12 | 0.66 | 0.04 | 25.74 | 0.17 | 0.15 |
| NHDF R2[a] | 0.93 | 2.94 | 2.40 | 1.11 | 0.58 | 0.04 | 25.36 | 0.13 | 0.15 |
| NHDF R3[a] | 0.91 | 3.41 | 3.14 | 1.15 | 0.64 | 0.04 | 30.00 | 0.14 | 0.12 |
| NHDF R4[a] | 0.88 | 3.33 | 2.88 | 1.21 | 0.51 | 0.04 | 28.03 | 0.12 | 0.14 |
| **mean** | **0.88** | **3.20** | **2.57** | **1.15** | **0.60** | **0.04** | **27.28** | **0.14** | **0.14** |
| **SE** | **0.03** | **0.10** | **0.28** | **0.02** | **0.03** | **0.00** | **1.08** | **0.01** | **0.01** |
| HSMC R1[a] | 1.56 | 2.40 | 17.04 | 1.57 | 0.66 | 0.15 | 17.06 | 0.36 | 0.48 |
| HSMC R2[a] | 1.18 | 2.54 | 15.87 | 1.40 | 0.47 | 0.14 | 17.76 | 0.28 | 0.31 |
| HSMC R3[a] | 1.14 | 2.39 | 10.88 | 1.97 | 0.43 | 0.12 | 16.00 | 0.20 | 0.21 |
| **mean** | **1.29** | **2.44** | **14.60** | **1.65** | **0.52** | **0.14** | **16.94** | **0.28** | **0.33** |
| **SE** | **0.13** | **0.05** | **1.89** | **0.17** | **0.07** | **0.01** | **0.51** | **0.04** | **0.08** |

second one was designed to amplify the region comprising the *SALNR*/AK091544 amplicon overlapping the XR_007061960 *HELLS* isoform (for abbreviation "*SALNR*" amplicon). The results of the qPCR showed a strong down-regulation of *HELLS* gene expression levels in the senescent cells compared to the young proliferating cells, thus confirming the *in silico* analysis. Similarly, the levels of *SALNR* were down-modulated in all the senescence cellular models (Table 3).

As expected, we found a strong correlation with a Pearson coefficient of 0.947 (Fig 4) between the expression levels measured with the qPCR amplicon amplifying the consensus region of major *HELLS* isoforms and those obtained with the *SALNR* amplicon overlapping the predicted XR_007061960 *HELLS* isoform (Table 3).

## Discussion

In this investigation, we applied an extensive *in silico* analysis of several publicly available RNA-seq datasets, including short- and long-read experiments, with the aim of disentangling the mystery of *SALNR*. We demonstrated that *SALNR*/AK091544 transcript is completely overlapped to a predicted *HELLS* transcriptional isoform and we did not find evidence of

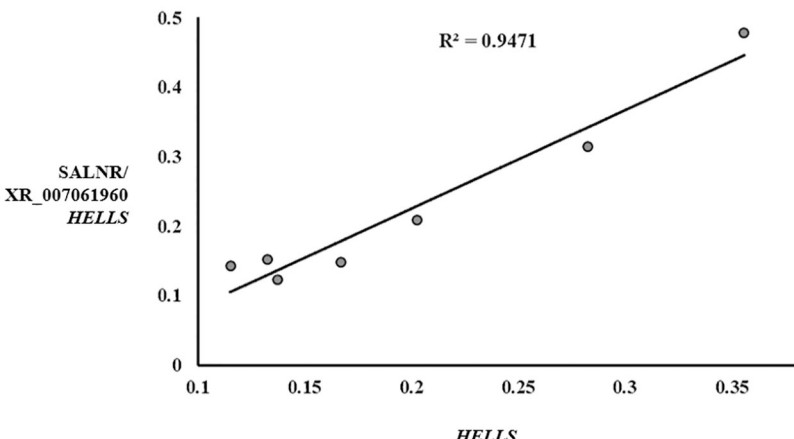

**Fig 4. Linear regression *HELLS*/*SALNR* gene expression.** Linear regression of gene expression levels (FC) of *HELLS* and *SALNR*/XR_007061960 *HELLS* amplicons of all senescent cells.

*SALNR*/AK091544 existence as independent transcript. By wet approaches (RT-PCR and Sanger sequencing), we confirmed the presence of this predicted *HELLS* isoform in human cells and tissues of different origin. Moreover, by re-analysis of RNA-seq datasets as well as by qPCR, we assessed the down-regulation of *HELLS* gene in old/senescent compared to young/proliferative human cells.

*SALNR* was first described by Wu et al. [8], who performed microarray experiments aimed at identifying deregulated lncRNAs during replicative senescence of lung fibroblasts and pointed to *SALNR*/AK091544 as a consistently down-regulated lncRNA in replicative senescence and in other cellular models of induced senescence. In our study, thanks to the availability of several publicly available RNA-seq datasets, we could deeply investigate the *HELLS* genomic region. Even if long-read datasets showed poor coverage of *HELLS* genomic portion, we found the presence of enough reads enabling a deep analysis. Our data are not necessarily at odds with the results shown by Wu and colleagues about *SALNR*. According to our results, two main hypotheses can be advanced. The first one supposes that *SALNR* does not exist as an independent transcriptional unit but it is a portion of the XR_007061960 *HELLS* isoform. Under this hypothesis, XR_007061960 should be considered as an *HELLS* sense-overlap lncRNA and the findings obtained by Wu and colleagues about *SALNR* could instead be attributed to the region of the XR_007061960 *HELLS* isoform overlapping *SALNR* rather than to the potential *SALNR* non-coding transcript. Failure to amplify the very long XR_007061960 transcript by RACE experiments could be caused by the intrinsic limitations of these techniques. For instance, 5'-RACE often yields truncated products when trying to obtain the amplification of regions longer than 1 Kb, due for example to the presence of secondary structures in the transcript that cause the detachment of reverse transcriptase. Of note, the XR_007061960 *HELLS* isoform overlapping *SALNR*/AK091544 was only added in the latest human genome annotation release of April 2022 and the RNA-seq datasets providing evidence on the existence of this isoform were all made available after the publication of Wu et al. paper. The second hypothesis is that both *SALNR* and the XR_007061960 *HELLS* isoform exist. In this case, the failure to detect *SALNR* transcripts could be caused by several reasons. Many lncRNAs may be particularly difficult to detect by analysis of steady-state RNA levels because they are expressed at very low levels and they are highly unstable. An example is provided by lncRNAs generated through premature transcription termination of protein-coding genes that are not polyadenylated and rapidly degraded [23]. Moreover, they often display a restricted tissue-specific or cell-specific expression pattern. In our study, we could not evaluate the existence of the XR_007061960 *HELLS* isoform in the cell line used by Wu and colleagues as model for cellular senescence. But despite the limitations discussed above, at least in the context of the experimental conditions adopted in our study, our findings do not support the existence of SALNR as a separate, independent transcript.

By *in silico* re-analysis of eight RNA-seq datasets (see Table 1) associated to transcriptomic investigation of replicative senescence in different human cells such as fibroblasts, smooth muscle cells, and endothelial cells, we found that the normalized read counts of *HELLS* were significantly down-regulated in senescent cells compared to young cells. In the same direction, by using *in vitro* cellular models of replicative senescence, we confirmed the down-regulation of *HELLS* in senescent NHDFs, as well as in HSMCs, compared to proliferating/young cells. What about the biological function of *HELLS*? Does *HELLS* play any role in senescence? *HELLS* gene (also known as *LSH*, *PASG*, *SMARCA6)* encodes for a helicase of 838 amino acids (Uniprot ID: Q9NRZ9) belonging to the SNF2 family of proteins and playing an essential role in normal development and involved in the regulation of lymphoid lineage expansion and survival. It is a chromatin remodeling protein with a role in the epigenetic regulation acting at different levels, including remodeling of nucleosomes, histone modifications, and *de novo* or

maintenance DNA methylation [24]. It has also been shown to have a role in double strand DNA repair [25]. It may play a role in the formation and organization of heterochromatin, implying a functional role in the regulation of transcription and mitosis [26]. Noteworthy, the relevance of epigenetic regulation in aging and senescence has increased by time [27]. Several pieces of evidence suggest the involvement of *HELLS* in senescence and aging: indeed, its expression is transcriptionally repressed by p53 via p21 and down-regulated during senescence [28]. Zhou et al. demonstrated that high *HELLS*/*Lsh* expression in young cells represses *p16INK4a* expression, thereby maintaining the proliferative state of 2BS cells, while loss of *Lsh* expression may lead to increased *p16INK4a* expression in senescent 2BS cells, contributing to the onset of cellular senescence [29,30]. In the same line, *Hells* knock-out mice display premature aging and the fibroblasts derived from these mice show a senescent phenotype [31]. Moreover, *HELLS* has been annotated in the database of genes related to aging, GenAge, with potential relevance to the human aging process.

## Conclusion

Overall, in this study, we demonstrated that *SALNR*/AK091544 sequence completely overlaps *HELLS* transcriptional isoforms; moreover, in the context of the experimental conditions adopted, it was not possible to validate the existence of *SALNR* as a separate, independent transcript in the human genome. This study indicates caution when working with genomic loci hosting lncRNA transcripts and suggests to adopt a deep assessment by *in silico* approaches before proceeding with further investigation on a specific transcript. Note that our investigation has revealed a coherent and fairly reproducible down-regulation of the *HELLS* gene in human cell models undergoing replicative senescence. Recently, the role of epigenetics in promoting senescence has been highlighted. In this scenario, the function of *HELLS* as an epigenetic regulator of senescence deserves further exploration.

## Supporting information

**S1 Fig. Short-read RNA-seq.** Visualization of sequence reads (from short-read RNA-seq) aligned to the genomic portion including *HELLS* and *SALNR*, using UCSC genome browser. It is not possible to isolate *SALNR* expression from this type of reads, but the heavy presence of reads covering the 3' end of the *HELLS* gene suggests an interesting production of known and novel isoforms in that location. This figure represents the sample with highest read count for each dataset (from proliferative cells). The GSE157867 track (the dataset with 2x150bp reads) is plotted in "squish" display mode, while the others are plotted in "dense" mode. The short-read RNA-seq mapping tracks can be visualized on UCSC genome browser: https://genome.ucsc.edu/s/cnr.itb.ba/short-read_RNA-seq. See Data Availability section.
(TIF)

**S2 Fig. XR_007061960 primers.** Genomic location of the two amplicons used to amplify the XR_007061960 isoform of *HELLS* and corresponding primer sequences.
(TIF)

**S3 Fig. β-gal staining.** Representative images of colorimetric β-gal assay in young (p6) (A) and old (p16) (B) HMSCs, and in young (p6) (C) and old (p16) (D) NHDFs (x10 magnification).
(TIF)

**S1 Table. Long-read RNA-seq dataset.** List of all long-read datasets that were mapped to the human genome to search for reads mapping to *HELLS* genomic location to identify the

isoform that originates them.
(XLSX)

**S2 Table. qPCR primers.** List of primers used for quantitative RT-PCR experiments.
(XLSX)

**S3 Table. RSEM and MultiDEA read count estimations.** Read count estimation computed by RSEM and MultiDEA for each isoform in *HELLS* genomic location, for each sample and for each dataset. Despite the diffuse overlapping among the isoforms, the Bayesian model included in RSEM tends to prefer only a small subset of isoforms, changing in each dataset. MultiDEA clearly shows that the only isoforms to achieve a significant number of uniquely mapping reads (>20) are XR_007061960 and ENST00000419900, while all the other reads map to more than one reference transcript. RSEM and MultiDEA columns list the isoform expression estimation computed by each tool; [A,B,C,D] is the multiread detail computed by MultiDEA (trapezoid vertices, A = uniquely mapping reads, B = reads having the isoform as unique best match; C = reads having that isoform as non-unique best match; D is the total number of reads that map to that isoform). Isoforms are sorted by start position on chr10; XR_007061960 is an additional isoform of *HELLS* proposed by RefSeq and ENST00000432120 is a lncRNA that overlaps *HELLS*.
(XLSX)

**S4 Table. DESeq2 on RSEM and MultiDEA expressions.** Results of DESeq2 differential expression analysis performed on RSEM and MultiDEA isoform expression estimation. The isoforms are sorted by start position on chr10, and even if RSEM and MultiDEA show some different results, it is possible to deduce that the senescence expression decrease influences both the transcript overlapping SALNR and the ones not overlapping it. Statistically significant count is the number of times that the isoform resulted significantly down-regulated in the 8 tested datasets; XR_007061960 is an additional isoform of *HELLS* proposed by RefSeq and ENST00000432120 is a lncRNA that overlaps *HELLS*; a result is considered statistically significant (*) if mean read count >20, absolute log2 Fold Change >1 and adjusted p-value (FDR) <0.05.
(XLSX)

**S1 File. Nucleotide sequences in FASTA format.** Results of Sanger sequencing carried out on RT-PCR products (HELLS_XR_007061960 and HELLS_XR_007061960_3' amplicons) obtained from HSMC and SH-SY5Y cells exported as sequences in FASTA format.
(FASTA)

**S1 Raw images.**
(PDF)

## Acknowledgments

We thank Paride Pelucchi, Elena Battaglioli, Emanuela Toffolo, Lisa Giovannelli for their help with setting cell culture models and for providing technical support. We thank Nicola Losito for maintaining the ICT infrastructure for bioinformatic analysis. SB is recipient of a scholarship of PhD in Experimental Medicine, Università degli Studi di Milano. CR is recipient of a scholarship of PhD in Pharmacological Biomolecular Sciences, Experimental and Clinical, Università degli Studi di Milano.

## Author Contributions

**Conceptualization:** Arianna Consiglio, Marco Venturin, Cristina Battaglia.

**Data curation:** Arianna Consiglio, Marco Venturin, Flavio Licciulli, Cristina Battaglia.

**Formal analysis:** Arianna Consiglio, Cristina Battaglia.

**Funding acquisition:** Arianna Consiglio, Marco Venturin, Cristina Battaglia.

**Investigation:** Arianna Consiglio, Marco Venturin, Sabrina Briguglio, Clara Rossi, Cristina Battaglia.

**Methodology:** Arianna Consiglio, Marco Venturin, Sabrina Briguglio, Cristina Battaglia.

**Project administration:** Marco Venturin, Cristina Battaglia.

**Resources:** Clara Rossi, Stefano Bellosta, Maria Grazia Cattaneo, Flavio Licciulli, Cristina Battaglia.

**Software:** Arianna Consiglio, Giorgio Grillo, Flavio Licciulli.

**Supervision:** Cristina Battaglia.

**Validation:** Arianna Consiglio, Marco Venturin, Cristina Battaglia.

**Visualization:** Arianna Consiglio, Marco Venturin, Cristina Battaglia.

**Writing – original draft:** Arianna Consiglio, Marco Venturin, Cristina Battaglia.

**Writing – review & editing:** Arianna Consiglio, Marco Venturin, Stefano Bellosta, Maria Grazia Cattaneo, Flavio Licciulli, Cristina Battaglia.

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
