## [Decision Letter · Decision Letter 0]

6 Mar 2023

PONE-D-23-03658Lost in HELLS: disentangling the mystery of SALNR existence in senescence cellular modelsPLOS ONE

Dear Dr. Battaglia,

Thank you for submitting your manuscript to PLOS ONE. After careful consideration, we feel that it has merit but does not fully meet PLOS ONE’s publication criteria as it currently stands. Therefore, we invite you to submit a revised version of the manuscript that addresses the points raised during the review process.

We look forward to receiving your revised manuscript.

Kind regards,

Jacopo Sabbatinelli, MD, PhD

Academic Editor

PLOS ONE

Journal Requirements:

6. Please upload a copy of Supporting Information Figure/Table/etc. S1,S2,S5, S6 Tables and S3,S4 Figures which you refer to in your text on page 19.

Reviewers' comments:

Reviewer's Responses to Questions

**Comments to the Author**

1. Is the manuscript technically sound, and do the data support the conclusions?

Reviewer #1: Partly

2. Has the statistical analysis been performed appropriately and rigorously? 

Reviewer #1: Yes

3. Have the authors made all data underlying the findings in their manuscript fully available?

Reviewer #1: Yes

4. Is the manuscript presented in an intelligible fashion and written in standard English?

Reviewer #1: Yes

5. Review Comments to the Author

Reviewer #1: The manuscript "Lost in HELLS: disentangling the mystery of SALNR existence in senescence cellular models" by Consiglio A. et al., tried highlighting the existence of SALNR, an aging-associated long non-coding RNA first described by Wu et al. in 2015. By both in silico and in vitro approaches, the Authors showed that the SALNR transcript is completely overlapped to a transcriptional isoform (XR_007061960) of HELLS, a gene encoding for a helicase belonging to the SNF2 family of proteins. The Authors validated the existence of the HELLS isoform by analysis of publicly available RNA-seq datasets and by RT-PCR experiments, but not the existence of SALNR lncRNA as a separate and independent transcript. This study seems interesting in the elucidation of the complex cellular senescent regulatory network involving HELLS and SALNR lncRNA. However, I have several points that authors could consider to improve this manuscript.

1) The study attempts to demonstrate that SALNR lncRNA does not exist as a separate and independent transcript, as previously reported by Wu et al. 2015, therefore claiming to "exclude SALNR from the list of lncRNAs associated with senescence". However, the data do not fully support this conclusion. This study is not intended to replicate the study by Wu et al. 2015, and accordingly, the experimental conditions are widely different. In particular, the rationale for choosing different cell lines (in place of WI-38 cell line) for RT-PCR analysis is unclear: different cell types have different transcriptomes. I agree it is interesting to see if, in a broader set of cells from different origins, the Authors could obtain the same results, but otherwise these are not necessarily at odds with the findings shown by Wu and colleagues. The Authors mentioned this concept in the discussion, but it contradicts the claim in the conclusions. Similarly, the Authors did not perform any investigations regarding the interaction of SALNR or XR_007061960 with NF90 RNA-binding protein, but they argued that “the robust and consistent findings that Wu and colleagues obtained about SALNR could instead be attributed to the region of the XR_007061960 HELLS isoform overlapping SALNR rather than to the potential SALNR noncoding transcript, including the ability of SALNR to delay senescence when overexpressed, the interaction with NF90 RNA-binding protein and its effect on NF90 nuclear localization”. For instance, if the Authors could perform the same analysis of Wu et al. 2015 on the same cell types with the same experimental conditions, it would certainly strengthen the observation, but this is beyond the scope of this study.

Therefore, I would suggest specifying that it was not possible to validate the existence of SALNR as a separate, independent transcript exclusively in the context of the experimental conditions adopted in the study. Discussion and conclusions sections should be accordingly rewritten.

2) The limitations of this study are not discussed. This scenario demands at least a brief elucidation of reasons that can lead to the failure of SALNR recognition. There are interesting relevant articles published recently, from those authors may get help. As an e.g., lncRNAs are often tissue- and cell-specific, expressed at low levels, and most non-polyadenylated lncRNAs are rapidly degraded. They generally reflect patterns of protein-coding gene expression in a particular tissue type. Protein-coding genes often form truncated transcripts through premature transcription termination, and these transcripts can be considered as categories of lncRNAs (PMID: 35079163). Moreover, since pseudogenes typically produce lncRNAs, the actual gene and the long non-coding transcript can be recognized using the same primers (PMID: 36625940, PMID: 32185172). Another difficulty arises when lncRNAs are expressed "sense-overlap" to a recognized protein-coding gene, such as in the case of SALNR and HELLS. Intriguingly, pseudogenes such as some lncRNAs have open reading frames and encode proteins or peptides (PMID: 34947885).

3) In Figure 2. “Alignment of long–reads RNA-seq data on HELLS genomic location”, the Authors show 3 representative samples of long-read RNA-seq and their alignment on HELLS genomic location. Also, they state that “none of the reads exactly matches the SALNR/AK091544 location starting and ending in its candidate boundaries”. It is unclear why they did not choose to show the reads in SRR14638806 run from SRP012412 SRA study, defined as “reads overlapping SALNR” in the first record of Table S1 (BioProject accession: PRJNA63443). Please explain this choice.

4) It is not easy to understand the bioinformatic methodology used in differential expression analysis of RNA-seq datasets. Please explain it better in the methods and/or provide the R-code or a markdown document. Specifically, could be of interest to the readers to know: i) the quality of the RNA-seq dataset analyzed (FastQC check results), ii) the normalization method used, and iii) how potential “batch effects” have been handled.

5) The Authors loaded the results of some ad hoc Bash scripts into the UCSC genome browser. Is it possible to insert the corresponding URL of the results, please? The URL https://genome.ucsc.edu/ provided is the genome UCSC browser home page.

Minor issues:

Please check the manuscript for phrasing problems, e.g.:

1) “the read in SRR12524789 from SRP278890 run confirms…” maybe should be “the read in SRR12524789 run from SRP278890 confirms…”. The same has been reported in the caption of Fig. 2.

6. PLOS authors have the option to publish the peer review history of their article (what does this mean?). If published, this will include your full peer review and any attached files.

Reviewer #1: No

---

## [Author Response · Author response to Decision Letter 0]

21 Apr 2023

Reviewer #1: The manuscript "Lost in HELLS: disentangling the mystery of SALNR existence in senescence cellular models" by Consiglio A. et al., tried highlighting the existence of SALNR, an aging-associated long non-coding RNA first described by Wu et al. in 2015. By both in silico and in vitro approaches, the Authors showed that the SALNR transcript is completely overlapped to a transcriptional isoform (XR_007061960) of HELLS, a gene encoding for a helicase belonging to the SNF2 family of proteins. The Authors validated the existence of the HELLS isoform by analysis of publicly available RNA-seq datasets and by RT-PCR experiments, but not the existence of SALNR lncRNA as a separate and independent transcript. This study seems interesting in the elucidation of the complex cellular senescent regulatory network involving HELLS and SALNR lncRNA. However, I have several points that authors could consider to improve this manuscript.

1) The study attempts to demonstrate that SALNR lncRNA does not exist as a separate and independent transcript, as previously reported by Wu et al. 2015, therefore claiming to "exclude SALNR from the list of lncRNAs associated with senescence". However, the data do not fully support this conclusion. This study is not intended to replicate the study by Wu et al. 2015, and accordingly, the experimental conditions are widely different. In particular, the rationale for choosing different cell lines (in place of WI-38 cell line) for RT-PCR analysis is unclear: different cell types have different transcriptomes. I agree it is interesting to see if, in a broader set of cells from different origins, the Authors could obtain the same results, but otherwise these are not necessarily at odds with the findings shown by Wu and colleagues. The Authors mentioned this concept in the discussion, but it contradicts the claim in the conclusions. 

Similarly, the Authors did not perform any investigations regarding the interaction of SALNR or XR_007061960 with NF90 RNA-binding protein, but they argued that “the robust and consistent findings that Wu and colleagues obtained about SALNR could instead be attributed to the region of the XR_007061960 HELLS isoform overlapping SALNR rather than to the potential SALNR noncoding transcript, including the ability of SALNR to delay senescence when overexpressed, the interaction with NF90 RNA-binding protein and its effect on NF90 nuclear localization”. For instance, if the Authors could perform the same analysis of Wu et al. 2015 on the same cell types with the same experimental conditions, it would certainly strengthen the observation, but this is beyond the scope of this study.

Therefore, I would suggest specifying that it was not possible to validate the existence of SALNR as a separate, independent transcript exclusively in the context of the experimental conditions adopted in the study. Discussion and conclusions sections should be accordingly rewritten.

Answer: we thank the reviewer for the comments and suggestions. As commented by the reviewer our study is not intended to replicate the study by Wu et al 2015 neither to investigate the interaction of SALNR or XR_007061960 with NF90 RNA-binding protein but rather it is aimed to address the question if that SALNR region is a part of the predicted non-coding XR_007061960 HELLS isoform and to validate the existence of this isoform.

Unfortunately, it is not possible to validate the existence of SALNR in the context of the specific cell line adopted in Wu et al study because neither the human fetal lung diploid fibroblasts 2BS cells nor microarray/RNAseq data are publicly available. The 2BS cell line was established by Chinese researchers and to our knowledge it is not commercially available.

By exploiting several publicly available RNA-seq datasets, we could verify the presence of the SALNR sequence in human cell lines and tissues of different origin, including human fetal lung diploid fibroblasts WI-38 also used by Wu et al 2015. Moreover, by using long-read RNA-seq datasets, we have deeply investigated the HELLS genomic region. However, it is true that it was not possible to confirm the existence of SALNR transcript exclusively in the context of the experimental conditions adopted in our study. As suggested by the reviewer we revised the discussion and conclusions sections taking into account this limitation.

2) The limitations of this study are not discussed. This scenario demands at least a brief elucidation of reasons that can lead to the failure of SALNR recognition. There are interesting relevant articles published recently, from those authors may get help. As an e.g., lncRNAs are often tissue- and cell-specific, expressed at low levels, and most non-polyadenylated lncRNAs are rapidly degraded. They generally reflect patterns of protein-coding gene expression in a particular tissue type. Protein-coding genes often form truncated transcripts through premature transcription termination, and these transcripts can be considered as categories of lncRNAs (PMID: 35079163). Moreover, since pseudogenes typically produce lncRNAs, the actual gene and the long non-coding transcript can be recognized using the same primers (PMID: 36625940, PMID: 32185172). Another difficulty arises when lncRNAs are expressed "sense-overlap" to a recognized protein-coding gene, such as in the case of SALNR and HELLS. Intriguingly, pseudogenes such as some lncRNAs have open reading frames and encode proteins or peptides (PMID: 34947885).

Answer: we thank the reviewer for the constructive revision and the suggestion of interesting articles on lncRNAs that helped us to integrate the discussion with a brief elucidation on the reasons that can lead to the failure of SALNR recognition.

We have revised the discussion as follows: 

“SALNR was first described by Wu et al. [8], who performed microarray experiments aimed at identifying deregulated lncRNAs during replicative senescence of lung fibroblasts and pointed to SALNR/AK091544 as a consistently down-regulated lncRNA in replicative senescence and in other cellular models of induced senescence. In our study, thanks to the availability of several publicly available RNA-seq datasets, we could deeply investigate the HELLS genomic region. Even if long-read datasets showed poor coverage of HELLS genomic portion, we found the presence of enough reads enabling a deep analysis. Our data are not necessarily at odds with the results shown by Wu and colleagues about SALNR. According to our results, two main hypotheses can be advanced. The first one supposes that SALNR does not exist as an independent transcriptional unit but it is a portion of the XR_007061960 HELLS isoform. Under this hypothesis, XR_007061960 should be considered as an HELLS sense-overlap lncRNA and the findings obtained by Wu and colleagues about SALNR could instead be attributed to the region of the XR_007061960 HELLS isoform overlapping SALNR rather than to the potential SALNR non-coding transcript. Failure to amplify the very long XR_007061960 transcript by RACE experiments could be caused by the intrinsic limitations of these techniques. For instance, 5’-RACE often yields truncated products when trying to obtain the amplification of regions longer than 1 Kb, due for example to the presence of secondary structures in the transcript that cause the detachment of reverse transcriptase. Of note, the XR_007061960 HELLS isoform overlapping SALNR/AK091544 was only added in the latest human genome annotation release of April 2022 and the RNA-seq datasets providing evidence on the existence of this isoform were all made available after the publication of Wu et al. paper. The second hypothesis is that both SALNR and the XR_007061960 HELLS isoform exist. In this case, the failure to detect SALNR transcripts could be caused by several reasons. Many lncRNAs may be particularly difficult to detect by analysis of steady-state RNA levels because they are expressed at very low levels and they are highly unstable. An example is provided by lncRNAs generated through premature transcription termination of protein-coding genes that are not polyadenylated and rapidly degraded [23]. Moreover, they often display a restricted tissue-specific or cell-specific expression pattern. In our study, we could not evaluate the existence of the XR_007061960 HELLS isoform in the cell line used by Wu and colleagues as a model for cellular senescence. But despite the limitations discussed above, at least in the context of the experimental conditions adopted in our study, our findings do not support the existence of SALNR as a separate, independent transcript.”

3) In Figure 2. “Alignment of long–reads RNA-seq data on HELLS genomic location”, the Authors show 3 representative samples of long-read RNA-seq and their alignment on HELLS genomic location. Also, they state that “none of the reads exactly matches the SALNR/AK091544 location starting and ending in its candidate boundaries”. It is unclear why they did not choose to show the reads in SRR14638806 run from SRP012412 SRA study, defined as “reads overlapping SALNR” in the first record of Table S1 (BioProject accession: PRJNA63443). Please explain this choice.

Answer: thank you for your comments. All the results of long-read RNA-seq data analysis were shown through the link in “Data availability”: https://genome.ucsc.edu/s/cnr.itb.ba/long-read_RNA-seq. We did not include SRR14638806 in Figure 2 because its reads only partly overlap SALNR as already shown for other SRRnnnn reads in the figure.

4) It is not easy to understand the bioinformatic methodology used in differential expression analysis of RNA-seq datasets. Please explain it better in the methods and/or provide the R-code or a markdown document. Specifically, could be of interest to the readers to know: i) the quality of the RNA-seq dataset analyzed (FastQC check results), ii) the normalization method used, and iii) how potential “batch effects” have been handled.

Answer: thank you for your comments. 

4.i) We have better explained the bioinformatic methodology used by including additional details in “Bioinformatic analysis” section:

“Below is a list of non-default parameters used for each tool. STAR was run through RSEM with --star option, and --paired-end was used for paired end samples, while deSALT was run with --gtf to use GTF annotation files and with -x clr for PacBio reads (PacBio SMRT CLR reads, error rate 15%) or -x ont1d for Nanopore reads (Oxford Nanopore 1D reads, error rate > 20%). A table containing read ID, transcript ID and mapping quality was extracted from the BAM output of mapping and used as input for MultiDEA’s module mapping_output_2_trapezoids.sh. DESeq2 was applied to raw read counts by testing design = ~ condition, where condition is “proliferative” or “senescent”. We applied DESeq2 at gene level to RSEM read counts and at isoform level to both RSEM and MultiDEA read counts, with the aim of correlating the differential expression events to the different isoforms of HELLS.”

4.ii) FastQC check results were loaded in the UNIMI Dataverse (University of Milan)

DOI ( https://doi.org/10.13130/RD_UNIMI/TFQF8T). We have added a sentence in the method section.

4.iii) We did not specify the normalization method used because DESeq2 automatically normalizes the values by applying the median of ratios method.

4.iv) Since the datasets that we collected come from different SRA projects performed with different end time definition (number of days considered as senescent), we preferred to keep them separate during the analyses. We assumed that a set of FASTQ files coming from the same SRA project was produced in the same batch (unless otherwise specified), and we compared proliferative and senescent cells within the same SRA project and the same cell line, only (no “batch effects” had to be handled). Table 2 shows the results obtained for each dataset.

5) The Authors loaded the results of some ad hoc Bash scripts into the UCSC genome browser. Is it possible to insert the corresponding URL of the results, please? The URL https://genome.ucsc.edu/ provided is the genome UCSC browser home page.

Answer: the corresponding URL of the results are listed in the “Data Availability” section. We specified it better also at the end of the "Bioinformatic analysis" section:

“The results were loaded into the UCSC genome browser [https://genome.ucsc.edu/], and the corresponding URLs are listed in the Data availability section.”

and in the Results:

“The results were loaded into the UCSC genome browser (genome release GRCh38/hg38, see Data availability for the corresponding URLs).”

Minor issues:

Please check the manuscript for phrasing problems, e.g.:

1) “the read in SRR12524789 from SRP278890 run confirms…” maybe should be “the read in SRR12524789 run from SRP278890 confirms…”. The same has been reported in the caption of Fig. 2.

Answer: thank you for the suggestion. The sentence was revised accordingly and the whole article was checked for phrasing problems.

---

## [Decision Letter · Decision Letter 1]

9 May 2023

Lost in HELLS: disentangling the mystery of SALNR existence in senescence cellular models

PONE-D-23-03658R1

Dear Dr. Battaglia,

We’re pleased to inform you that your manuscript has been judged scientifically suitable for publication and will be formally accepted for publication once it meets all outstanding technical requirements.

Kind regards,

Jacopo Sabbatinelli, MD, PhD

Academic Editor

PLOS ONE

Additional Editor Comments (optional):

Reviewers' comments:

Reviewer's Responses to Questions

**Comments to the Author**

1. If the authors have adequately addressed your comments raised in a previous round of review and you feel that this manuscript is now acceptable for publication, you may indicate that here to bypass the “Comments to the Author” section, enter your conflict of interest statement in the “Confidential to Editor” section, and submit your "Accept" recommendation.

Reviewer #1: All comments have been addressed

2. Is the manuscript technically sound, and do the data support the conclusions?

Reviewer #1: Yes

3. Has the statistical analysis been performed appropriately and rigorously? 

Reviewer #1: Yes

4. Have the authors made all data underlying the findings in their manuscript fully available?

Reviewer #1: Yes

5. Is the manuscript presented in an intelligible fashion and written in standard English?

Reviewer #1: Yes

6. Review Comments to the Author

Reviewer #1: The study "Lost in HELLS: disentangling the mystery of SALNR existence in senescence cellular models" by Consiglio A. et al., represents an interesting topic in the elucidation of the complex cellular senescent regulatory network involving HELLS and SALNR lncRNA.

In the revised version of the manuscript, all concerns raised during the initial review process have been adequately addressed. The methodology is now better described and more transparent. The results are clearly presented and contextualized in the discussion. The expanded discussion section provides a more comprehensive interpretation of the results, and it now discusses the limitations of the study and the potential implications of the findings for future research in the field. Overall, the revised manuscript has been improved, and I believe it is now suitable for publication in PLOSONE Journal.

7. PLOS authors have the option to publish the peer review history of their article (what does this mean?). If published, this will include your full peer review and any attached files.

Reviewer #1: No

---

## [Editor Report · Acceptance letter]

19 May 2023

PONE-D-23-03658R1 

Lost in *HELLS*: disentangling the mystery of *SALNR* existence in senescence cellular models 

Dear Dr. Battaglia:

I'm pleased to inform you that your manuscript has been deemed suitable for publication in PLOS ONE. Congratulations! Your manuscript is now with our production department. 

Kind regards, 

on behalf of

Dr. Jacopo Sabbatinelli 

Academic Editor

PLOS ONE